# mRNA and miRNA Expression Analysis Reveal the Regulation for Flower Spot Patterning in *Phalaenopsis* ‘Panda’

**DOI:** 10.3390/ijms20174250

**Published:** 2019-08-30

**Authors:** Anjin Zhao, Zheng Cui, Tingge Li, Huiqin Pei, Yuhui Sheng, Xueqing Li, Ying Zhao, Yang Zhou, Wenjun Huang, Xiqiang Song, Ting Peng, Jian Wang

**Affiliations:** 1Key Laboratory of Ministry of Education for Genetics and Germplasm Innovation of Tropical Special Trees and Ornamental Plants, Hainan Key Laboratory for Biology of Tropical Ornamental Plants Germplasm, College of Forestry, Hainan University, Haikou 570228, China; 2Research Center for Terrestrial Biodiversity of the South China Sea, College of Forestry, Hainan University, Haikou 570228, China; 3Department of Development and Design, Hainan University, Haikou 570228, China; 4Key Laboratory of Germplasm Innovation on Protection and Conservation of Mountain Plant Resources, Ministry of Education/College of Agriculture, Guizhou University, Guiyang 550025, China

**Keywords:** *Phalaenopsis*, transcriptome, microRNA, anthocyanin biosynthesis, molecular mechanism

## Abstract

*Phalaenopsis* cultivar ‘Panda’ is a beautiful and valuable ornamental for its big flower and unique big spots on the petals and sepals. Although anthocyanins are known as the main pigments responsible for flower colors in *Phalaenopsis*, and the anthocyanins biosynthetic pathway in *Phalaenopsis* is generally well known, the detailed knowledge of anthocynins regulation within the spot and non-spot parts in ‘Panda’ flower is limited. In this study, transcriptome and small RNA libraries analysis from spot and non-spot sepal tissues of ‘Panda’ were performed, and we found *PeMYB7*, *PeMYB11*, and miR156g, miR858 is associated with the purple spot patterning in its sepals. Transcriptome analyses showed a total 674 differentially expressed genes (DEGs), with 424 downregulated and 250 upregulated (Non-spot-VS-Spot), and 10 candidate DEGs involved in anthocyanin biosynthetic pathway. The qPCR analysis confirmed that seven candidate structure genes (*PeANS*, *PeF3′H*, *PeC4H*, *PeF3H*, *PeF3H1, Pe4CL2*, and *PeCHI*) have significantly higher expressing levels in spot tissues than non-spot tissues. A total 1552 differentially expressed miRNAs (DEMs) were detected with 676 downregulated and 876 upregulated. However, microRNA data showed no DEMs targeting on anthocyanin biosynthesis structure gene, while a total 40 DEMs target transcription factor (TF) genes, which expressed significantly different level in spot via non-spot sepal, including 2 key MYB regulator genes. These results indicated that the lack of anthocyanidins in non-spot sepal may not directly be caused by microRNA suppressing anthocyanidin synthesis genes rather than the MYB genes. Our findings will help in understanding the role of miRNA molecular mechanisms in the spot formation pattern of *Phalaenopsis*, and would be useful to provide a reference to similar research in other species.

## 1. Introduction

Flower spots are heterochromatic dots or streaks with a specific texture and pattern appearing on the corollas of plants, which can affect the behavior of pollinators and the ornamental value of flowers [1,2,3,4]. Previous studies have confirmed the flower spot is caused by the accumulation of anthocyanins in a specific area of corollas. For example, peonidin-3-O-glucoside, malvidin-3-*O*-glucoside, delphinidin-3-*O*-diglucoside, and cyanidin-3-*O*-glucoside are the main anthocyanins found in petal and sepal spots in *Oncidium* [5], and cyanidin and delphinidin are the main anthocyanidins in the spot of pansy (*Viola × wittrockiana* Gams.) petals [6].

Molecular mechanism of anthocyanins accumulation has been clearly studied in some plants, such as *Dianthus hybrida* [7], *Antirrhinum majus* [8], *Petunia hibrida* [9], *Platycodonis Radix* [10], *Gerbera jamesonii* [11], and *Phalaenopsis equestris* [12], and the anthocyanin pathway, which is a branch of the flavonoid pathway, has been elucidated as well [13,14]. Three kinds of transcription factor genes families—MYB transcription factor, bHLH transcription factor, and WD40 repeat protein family (MBW)—were also found to regulate the anthocyanin biosynthesis genes [15].

As for the accumulation of anthocyanin in specific regions of colloras, the direct cause attributes to the specific expression of the biosynthesis genes involved in the anthocyanin pathway. For example, high level expression of the genes of OgCHI and OgDFR results in anthocyanin accumulation and pigmented spot formation in yellow lip in *Oncidium* [5,16]. Upregulation of *LhCHSA, LhCHSB*, and *LhDFR* is detected within the spots located in the center of the petals, in comparison to the low expression levels in the margin in *Lilium* ‘Sorbonne’ [17]. In *Dendrobium moniliforme*, pigment accumulation in the base of the column has been caused by a consequence of preferential expression of *DmF3′5′H* [18]. In *Clarkia gracilis*, precise spatiotemporal regulation of the expression of the anthocyanin genes *F3′H*, *F3′5′H*, *DFR1*, and *DFR2* produces spotted petals [19]. In pansy, *VwDFR*, *VwF3′5′H* and *VwANS* have more significantly higher level expression in cyanic flower areas [6]. Moreover, the MYB genes also play an important role for the production of flower spot by regulating the anthocyanin biosynthesis genes. For example, the large purple spots in *Phalaenopsis* ‘Everspring Fairy’ was mainly caused by the expression of *MYB* [12], and the *LhMYB6* and *LhMYB12* positively regulate anthocyanin biosynthesis and determine organ- and tissue-specific accumulation of anthocyanin in Asiatic hybrid lily ‘Montreux’ [20]. Hsu also conducted a detailed study on three *MYBs* in *Phalaenopsis*, and found that the color patterning of flower sepals, petals and lips is regulated by different MYB genes combinations, and the pigmented veins and spots on the petals are also regulated by these three *MYB* genes [21].

RNA interference, regulating gene expression by post-transcriptional mechanisms, has also been recognized to play an important role in the color special patterning model of some plants [22]. Koseki et al. [23] found that the star-type color pattern of *Petunia hybrida* Red star’ flowers is induced by sequence-specific degradation of chalcone synthase RNA. Their further study found that the formation of bicolor flower types of petunia was due to RNA interference in two *CHSA* alleles (*PhCHS-A1* and *PhCHS-A2*) [23]. In *Arabidopsis thaliana*, miRNAs may function as regulators in anthocyanin biosynthesis by targeting on related transcription factors and lead to the different accumulation of anthocyanin [24].

*Phalaenopsis* spp. have become important ornamental plants worldwide for their long-lasting and various colorful flowers [25]. There are varieties of flower colors and corolla pigmentation patterning styles of *Phalaenopsis* spp., so it is very important to find more details about the flower color patterning for its breeding and production. However, there was little research in this field because of the complex anthocyanin synthesis pathways and the gene expression networks in *Phalaenopsis* spp. RNA sequencing can effectively identify the subtle differences in gene expression and find the targeted genes of small RNA in transcription level, so it has been used to study flower color patterning recently, such as in *Lilium* ‘Tiny Padhye’ [26], monkeyflowers (*Mimulus*) [27], and tree peony [28]. In this study, the transcriptome sequencing and small RNA sequencing in spot and non-spot sepal of *Phalaenopsis* ‘Panda’ were carried out by genomics Illumina sequencing platform, and the expression levels of candidate genes, as well as microRNAs, were verified by qPCR. Meanwhile, the model of flower spot formation pattern was predicted by the interactions of the structural genes, regulatory genes together with small RNA. This study created a joint research of transcriptome sequencing and small RNA sequencing to explain the spot pigmentation in *Phalaenopsis.* spp., and the results will be helpful for the breeding of new colorful cultivars of *Phalaenopsis* spp.

## 2. Results

### 2.1. Anthocyanin Accumulation Patterns in Phalaenopsis ‘Panda’

The examination by light microscope showed that anthocyanin accumulated in the upper epidermal cells of spot area, while no anthocyanin accumulated in cells in non-spot area (Figure 1A–C). This result indicated that the visible spots resulted from accumulation of anthocyanin in cells of the upper epidermis. Furthermore, the anthocyanin content of the spot in *Phalaenopsis* ‘panda’ sepals was significantly different from that of the sepal according to the results by the UV-visible spectrophotometer scanning detection (Figure 1D,E). Anthocyanin content measured by the pH differential method revealed that high content of anthocyanin (1.0798 mg/g) was accumulated in spot area, while was barely detectable in the acyanic parts extraction.

### 2.2. Construction of cDNA Library and Gene Mapping to the Reference Genomes

Four cDNA libraries were constructed using Illumina Hiseq 2000 platform (Illumina, San Diego, CA, USA) and 44,632,028, 45,110,536, 45,380,086, 44,434,116 high-quality reads were obtained, respectively (Appendix A). The sequencing raw data has been deposited into the Short Reads Archive (SRA) database under the accession number SRP166213. These clean reads were mapped to reference genome of *Phalaenopsis equestris* and the average gene mapping ratio of each sample was 55.47%. We considered that the low homology between *Phalaenopsis* ‘Panda’ and *Phalaenopsis equestris* result to the low mapping ratio, but it has no effect on RNAseq quantitative analysis owing to the high clean reads quantity and enough sequencing data (Appendix A).

### 2.3. Functional Annotation and Classification

To annotate the gene with putative functions, the assembled genes were searched against the public databases of NR. Among them, 17,871 genes were annotated to the NR database. To further illustrate the main biological functions of the transcripts, GO (15,588 genes) and KEGG pathway (19,864 genes) analyses were performed (Appendix A).

### 2.4. Identification of Differentially Expressed Genes and KEGG Enrichment Analysis of DEGs

A total of 674 DEGs were detected in pairwise comparison with 424 downregulated and 250 upregulated genes (Non-spot-VS-Spot) (Figure 2). A total of 543 DEGs were mapped to all 106 pathways. Notably, the Flavonoid biosynthesis (ko00941, 12 DEGs of 105 gene with *Q-*value = 0.0005279107) and Phenylalanine metabolism (ko00360, 8 DEGs of 132 gene with *Q-*value = 1.900591 × 10^−1^) were most significantly enriched in top20 pathways (Figure 3, Appendix A).

### 2.5. DEGs in Anthocyanin Biosynthesis and MBW Genes

There were 10 DEGs in flavonoid biosynthesis which are directly related to spot pattern, including *PeANS* (PEQU_25924), *PeF3′H* (PEQU_00400), *PeC4H* (PEQU_12025), *PeDFR* (PEQU_34933), *PeCHI* (PEQU_22606), *PeF3H1* (PEQU_38891), *PeF3H* (PEQU_22432), *PePAL* (PEQU_01877), *Pe4CL2* (PEQU_00756), Pe4CL (PEQU_07458), and most of them presented higher expression level in spot areas than non-spot areas (Table 1, Figure 4).

As for the MBW transcriptor genes related to the anthocyanin synthesis, 4 unigenes were found having significant expression difference including 3 MYB uingenes and 1 bHLH unigene (Table 2). In these transcriptor unigenes, 2 MYB unigenes and 1 bHLH unigene were upregulated and 1 MYB unigene was downregulated.

### 2.6. Data Analysis of Small RNA Sequencing

Four sRNA libraries were constructed and 23,352,936, 24,896,769, 26,605,438, 25,926,370 high-quality tag were obtained, respectively (Appendix A). The sequencing raw data has been deposited into the Short Reads Archive (SRA) database under the accession number SRP161646. These clean tags were mapped to reference genome of *Phalaenopsis equestris* and other sRNA databases. The average gene mapping ratio of each sample was 77.38% (Appendix A).

A total of 1552 DEMs were detected in pairwise comparison with 676 downregulated and 876 upregulated (Non-spot-VS-Spot) (Figure 5). miRNA target gene prediction resulted in 1396 common unigenes between TargetFinder (http://http://targetfinder.org/) and psRobot (http://http://omicslab.genetics.ac.cn/psRobot/), 1648 by TargetFinder and 7652 by psRobot, and totally 1307 target unigenes were aligned to KEGG database. However, there were no DEMs were found targeting on anthocyanin biosynthesis structure gene. As for the regulator genes, we found totally 12 microRNAs targeting on 10 significant different expressed MYB, *bHLH* or *WRKY* unigenes (Table 3). However, only *PeMYB7* (PEQU_03393), *PeMYB11* (PEQU_10361, PEQU_10362) were significantly upregulated in these TF families according to the transcriptomes data, and these microRNAs belonged to miR156 or miR858 families.

### 2.7. qPCR of Key Structural Genes, Regulate Genes and miRNA

The expression levels of 10 anthocyanin genes and 4 transcriptor genes between the non-spot and spot areas were verified by qPCR. The results showed the genes *Pe4CL2*, *PeANS*, *PeF3H*, *PeF3H1*, *PeF3′H* and *PeMYB7*, *PeMYB11* presented significantly higher expression levels in spot areas than the non-spot areas (Figure 6), which was generally consistent with the results of the transcriptome data (Table 1).

The expression of microRNA of RNA miR858, miR156g were also analyzed by qPCR between the non-spot and spot areas (Figure 7). The results showed that they presented higher expression levels in non-spot areas, which means their targeting genes, *PeMYB7* and *PeMYB11*, having less expression levels in the non-spot areas.

## 3. Discussion

### 3.1. Low Expression of Anthocyanin Genes Causing the Lack of Pigments in Non-Spot Areas

In this study, we found anthocyanin accumulated in the upper epidermal cells in spot area, while no anthocyanin accumulated in cells in non-spot areas of ‘Panda’ (Figure 1). Transcriptome data and qPCR indicated 7 anthocyanin genes were low expression in the non-spot areas, especially for the very different expression levels of *4CL2, F3’H*, *F3H* and *F3H1* (Table 2, Figure 6). In *Phalaenopsis* ‘Everspring Fairy’, which also has purple spot on the white petal and sepal, DFR is the main gene which differently expressed between the purple and white part of the petals and sepals [12]. However, in *Phalaenopsis* ‘Panda’ expression of *DFR* in spot tissue was not significantly changed in comparison to non-spot area. These results showed that the same phenotype may be caused by different genes in the pigmentation in *Phalaenopsis.*

### 3.2. PeMYB7 and PeMYB11 Are Important Genes in Spot Formation

MYBs have been found to be very important in the floral pigmentation patterning in *Phalaenopsis* spp. In the sepals/petals, silencing of *PeMYB2*, *PeMYB11*, and *PeMYB12* resulted in the loss of the full-red pigmentation, red spots, and venation patterns, respectively; *PeMYB11* was responsive to the red spots in the callus of the lip, and *PeMYB12* participated in the full pigmentation in the central lobe of the lip [21]. In *P. amabilis* and *P. schilleriana*, anthocyanin-specific Myc and Wd were expressed, however, Myb specific for anthocyanin biosynthesis were undetectable in *P. amabilis*; in *Phalaenopsis* ‘Everspring Fairy‘ petals and sepals, high levels of anthocyanin-specific Myb transcripts were present in the purple, but not in the white sectors [12]. Hsu, et al. [29] verified *PeMYB11* as the major regulatory R2R3-MYB transcription factor for regulating the production of the black color, and a retrotransposon, named Harlequin Orchid RetroTransposon 1 (HORT1), resulted in strong expression of *PeMYB11* and thus extremely high accumulation of anthocyanins in the harlequin flowers of the *P.* Yushan Little Pearl variety.

In our study, *PeMYB7* and *PeMYB11* expressed significantly different between the spot and non-spot areas, while *PeMYB2* and *PeMYB12* had not different expression levels (Table 2 and Figure 6). This result indicated that the gene of *PeMYB11* may also play an important role in spot pigmentation in *Phalaenopsis* ‘Panda’, which was similar to these previous researches [12,21,29,30]. *PeMYB7* had different expression levels between the spot and non-spot areas of sepals in this research, which indicated this gene may also associate with purple spot patterning in *Phalaenopsis* ‘Panda’. The function of *PeMYB7* is not very clear presently, however, in the phylogenetic tree inferred from MYB genes of *Phalaenopsis equestris* and *Oryza sativa* (Figure 8A), *PeMYB7* was in the same clad of *PeMYB2* and *PeMYB8*, suggesting *PeMYB7* may have similar function as *PeMYB2* which can activate anthocyanin synthesis [30].

### 3.3. miR156g, miR858 Silence PeMYB7, and PeMYB11

MiRNAs can influence tissue anthocyanin formation in previous studies. In petunia, the sequence-specific degradation of CHS RNA is the primary cause of the formation of white sectors in ‘Red Star’ flowers [23]. In potato (*Solanum tuberosum* L.), small RNAs of miR828 and TAS4 D4(−) can decrease the expression levels of MYB12 and R2R3-MYB genes in purple skin and flesh, which caused to anthocyanin accumulation [31]. In our research, miR156 and miR858 were predicted as the main interference RNAs for PeMYB7 and PeMYB11 (Table 3). In order to confirm *PeMYB7* and *PeMYB11* are the direct targets of the small RNAs, we download the complete cDNAs of *PeMYB7* (GenBank: KF769472.1) and *PeMYB11* (GenBank: MH675649.1), and analyzed the target sites of miR156 and miR858 in these two genes (Figure 8B). This finding confirms that miR156g and miR858 can silence *PeMYB7* and *PeMYB11*.

In *Arabidopsis thaliana*, miR156 negatively regulates the anthocyanin accumulation by regulating the expression level of target gene *MYB113* indirectly [32]. High miR156 level promotes anthocyanin biosynthesis through the negative regulation of *SPL9* gene because SPL9 can destabilize the MYB-bHLH-WD40 transcriptional activation complex. However, in this study, *SPL9* gene in *Phalaenopsis* doesn’t express differently, which indicated that mtr-miR156g-3p may be able to directly silence *PeMYB7* and caused the anthocyanin lacking on the white sector.

The biological functions of miR858 has not been fully explored, however, small RNA sequencing in *Arabidopsis thaliana* [24] and *Vitis vinifera* L. [33] verified that its target gene is *MYB12*, which is a positive transcript factor for the anthocyanin synthesis. Chitwood et al. transferred the constructed binary vector into wild-type tomato plants to reduced MIR858 expression level, and found the expression levels of *SLY-Myb12* in transgenic plants were lower than wild type, with upregulations of the related structure gene *PAL*, *DFR*, *ANS* and *CHS* involved in anthocyanin biosynthesis, and increasing of anthocyanin content [34]. Ballester et al. [35] used VIGS technology silenced *MYB12* and then found the tomato fruits turned pink. In our study the targeted genes of miR858 were predicted to be both *MYB12* and *MYB11* (Table 3), however, the unigene of *PeMYB12* didn’t show significantly different expression levels between the spot and non-spot areas, suggesting that miR858 explored higher effect on the unigene of *PeMYB11* in ‘Panda’.

### 3.4. A Proposed Modelsummarizing of Spot Formation Pattern in Phalaenopsis ‘Panda’

Our research confirmed *Pe4CL2*, *PeF3H*, *PeF3′H*, and *PeANS* expression were low or repressed in non-spot areas in ‘Panda’ sepals, and MIR156g and MIR858 may silence the expression of *PeMYB11* and *PeMYB7* in non-spots parts. In conclusion, we illustrated a proposed molecular mechanism in spots formation in ‘Panda’ (Figure 9). The diagram revealed the unique spot formation in ‘panda’ may result from the higher expression of MIR156g and MIR858 clusters in non-spot tissue, which targeted and suppressed the expression of key regulate genes *PeMYB7, PeMYB11* and then caused very low expression of *Pe4CL2*, *PeF3H*, *PeF3′H*, and *PeANS* in non-spot tissue. These results finally led to the lack of synthesis and accumulation of anthocyanin in non-spot area.

## 4. Materials and Methods

### 4.1. Plant Materials

*Phalaenopsis* ‘Panda’ (Figure 10) were grown at the horticultural farm of Hainan University (Latitude: 20.03N, longitude: 110.33E), Haikou, Hainan Province, China. The sepal tissues were divided into two parts, spots area and non-spots area for total anthocyanin analyses and totally RNA extraction. Three independent biological replicates were collected for each sample. All samples were immediately frozen in liquid nitrogen and stored at −80 °C until use.

### 4.2. Observations of Sepal Anatomy and Determination of Total Anthocyanin Content

Spot and non-spot sepal tissue of full bud were cut into segments longitudinally (approx.10 mm × 5 mm), and the upper epidermis and dorsal epidermis were separated with forceps. The sections were observed and photographed with a light microscope (Primo Vert, Zeiss, Germany).

The total anthocyanin content of *Phalaenopsis* sepal in spot and non-spot areas were determined by the pH differential method [36].

### 4.3. RNA Extraction, cDNA Library Construction, and mRNA Sequencing

RNAs were extracted from spot and non-spot parts of sepals of full bud using the modified Trizol method [6]. Two biological replicates were used for spot sepal and non-spot sepal. A total of 4 RNA-seq libraries were constructed from these sepals. RNA-Seq libraries were constructed using the RNA Library Prep Kit for Illumina according to the manufacturer’s instructions (NEB, Ipswich, MA, USA). Library quality was assessed on the Agilent Bioanalyzer 2100 system. The mRNA libraries were sequenced on the Illumina HiSeq 2000 platform (Illumina, San Diego, CA, USA) based on sequencing by synthesis with 150 bp paired-end reads (Biomarker Technologies, Wuhan, China).

### 4.4. mRNA Transcriptome Data Analysis

The reads with adaptors were removed firstly. The reads in which unknown bases comprised more than 5% of the total and low quality reads (the percentage of the low quality bases of quality value ≤ 15 is more than 20% in a read) were also removed. The clean reads were aligned to *Phalaenopsis equestris* genome accessed from (https://www.ncbi.nlm.nih.gov/bioproject/192198) by HISAT2 (http://www.ccb.jhu.edu/software/hisat/index.shtml) [37].

Gene alignment rate was counted with Bowtie2 v2.2.5 (http://bowtie-bio.sourceforge.net/bowtie2/index.shtml), gene and transcript expression levels were calculated with RSEM v1.2.12 (http://deweylab.biostat.wisc.edu/RSEM). The differentially expressed genes (DEGs) between spot and non-spot tissues were identified and filtered with the R package DESeq2 based on |log2 (foldchange)| ≥ 1 and Padj < 0.05 [38]. The heatmap displays of the Trimmed Mean of M-values (TMM) normalized against the Fragment Per Kilobase of transcripts per Million (FPKM) mapped reads were performed using the R package pheatmap (https://cran.r-project.org/src/contrib/Archive/pheatmap/). The DGEs were imported into Blast2GO software v2.5 [39] and in-house perl scripts for gene ontology (GO) term analysis, while the KEGG pathways were assigned to the assembled genes using the online KEGG Automatic Annotation Server (KAAS, http://www.genome.jp/kegg/), and then the phyper function in R software was used for enrichment analysis with *FDR* ≤ 0.01. To identify transcription factors, the assembled transcriptome were searched against the Plant Transcription Factor Database PlnTFDB (http://plntfdb.bio.uni-potsdam.de/v3.0/downloads.php) using hmmseach v3.0 (http://hmmer.org).

### 4.5. qRT-PCR Analyses of mRNA

10 DEGs in anthocyanin biosynthesis and 4 DEGs of transcription factor were chosen for qRT-PCR analyses. qRT-PCR analyses were performed using SYBR Premix Ex TaqTM II (Tli RNaseH Plus) (Takara, Dalian, China) according to the manufacturer’s instructions with the following reaction conditions: denaturation at 95 °C for 30 s and 40 cycles of amplification (95 °C for 5 s, 60 °C for 30 s). The β-Actin gene was used as an internal control for normalization. Relative expression levels of target genes were calculated using the 2^−∆∆*C*t^ [40] against the internal control. The gene-specific primers are shown in Appendix A. Experiments were performed with three independent biological replicates and three technical replicates.

### 4.6. Small RNA Library Construction and Sequencing

RNA was extracted from spot and non-spot sepal using the modified Trizol method [6], and 4 small RNA libraries from sepal spot and non-spot samples with 2 biological replicates were constructed and sequenced by SBQ500 sequencing method (BGI, Wuhan, China). Then the low-quality, 5′ primer contaminants, without 3′ primers and insert tags, and sequences fewer than 18 nucleotides (nt) were filtered out to obtain clean reads from raw data reads. The final clean reads of the 4 libraries were mapped to *Phalaenopsis* genome and other smallRNA databases using Bowtie2 [41]. All the remaining clean sequences were annotated into different classes to remove rRNA, scRNA, snoRNA, snRNA, and tRNA using miRBase, siRNA, piRNA, snoRNA database with Bowtie2 [42] and Rfam database with cmsearch [43]. The novel miRNAs were predicted by Mireap software (https://sourceforge.net/projects/mireap/).

### 4.7. Data Analysis of Small RNA Sequencing

The expression level of miRNAs was compared between spot and non-spot tissues to identify differentially expressed miRNAs (DEMs). The differentially expressed miRNA between spot and non-spot tissues were identified and filtered with the R package DESeq2 and Poisson Distribution [38,44]. The DEMs were determined based on FDR ≤ 0.001 and the absolute value of |Log2FC| ≥ 1. The heatmap displays of the Trimmed Mean of M-values (TMM) normalized against the Fragment Per Kilobase of transcripts per Million mapped reads (FPKM) were performed created using the R package pheatmap (https://cran.r-project.org/src/contrib/Archive/pheatmap/).

The potential target genes by differentially expressed miRNAs were predicted and analyzed with 2 different software, including psRobot [45] and Targetfinder [46]. In order to increase the level of confidence and get more reliable results, we selected only those binding sites that were predicted by both of two softwares.

All protein-coding target genes regulated by DEMs were annotated against the KEGG databases. The KEGG enrichment analysis for the target genes of the DEMs was performed by Fisher’s exact test (P-values) in Blast2GO pipeline, and P-values were used to conduct multiple test correction by FDR. KEGG terms with FDR< 0.05 were considered to be significantly enriched.

### 4.8. Verification of miRNA Expression Levels by qPCR

Two miRNA targets on key regulate gene transcription factor *PeMYB11* and *PeMYB7* were chosen for qRT-PCR analyses. The qRT-PCR analyses were performed using SYBR Premix Ex TaqTM II (Tli RNaseH Plus) (Takara, Dalian, China) according to the manufacturer’s instructions with the following reaction conditions: Denaturation at 95 °C for 30 s and 40 cycles of amplification (95 °C for 5 s, 60 °C for 30 s). The U6 gene was used as an internal control for normalization. Relative expression levels of target genes were calculated using the 2^−∆∆*C*t^ [40] against the internal control. The specific stem-ring primers were designed according to the miRNA qPCR primer design method [47], showed in Appendix A. Experiments were performed with three independent biological replicates and three technical replicates.

## Figures and Tables

**Figure 1 ijms-20-04250-f001:**
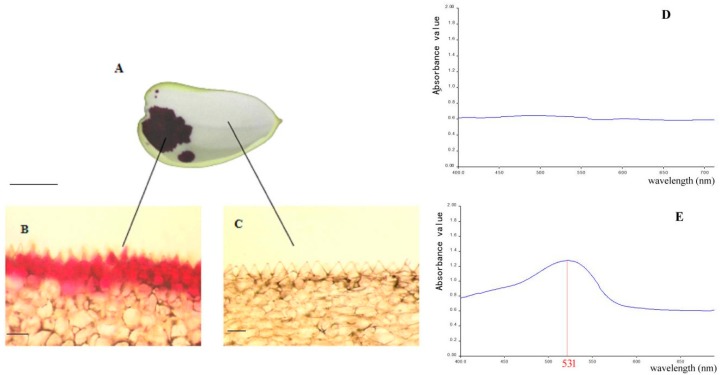
Anatomical structures and anthocyanin content of *Phalaenopsis* ‘Panda’ sepal. (**A**) Sepal (bar = 1 cm); (**B**) Upper epidermal cell in spot area (bar = 10 μm); (**C**) Upper epidermal cell in non-spot area (bar = 10 μm); (**D**) UV-visible spectrophotometer scanning of non-spot area; (**E**) UV-visible spectrophotometer scanning of spot area. Red number of 531 nm represents the absorption wavelength of anthocyanin.

**Figure 2 ijms-20-04250-f002:**
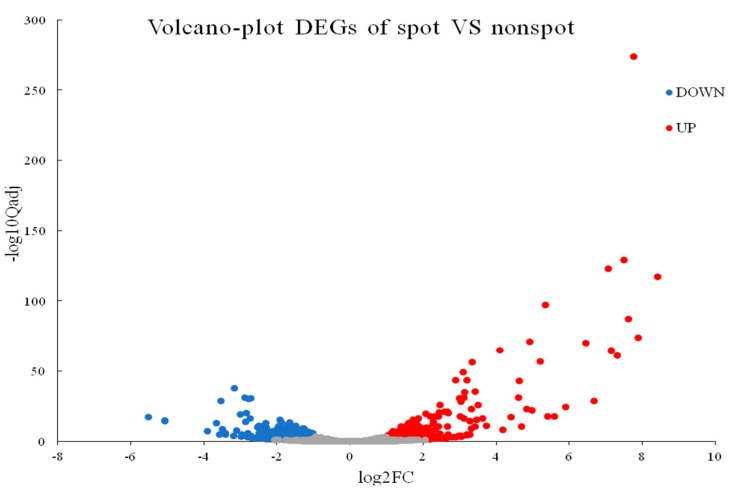
Volcano plot of differentially expressed genes (DEGs). X axis: log2 transformed fold change; Y axis: −log10 transformed significance; Red points: upregulated DEGs; Blue points: downregulated DEGs. Gray points: non-DEGs.

**Figure 3 ijms-20-04250-f003:**
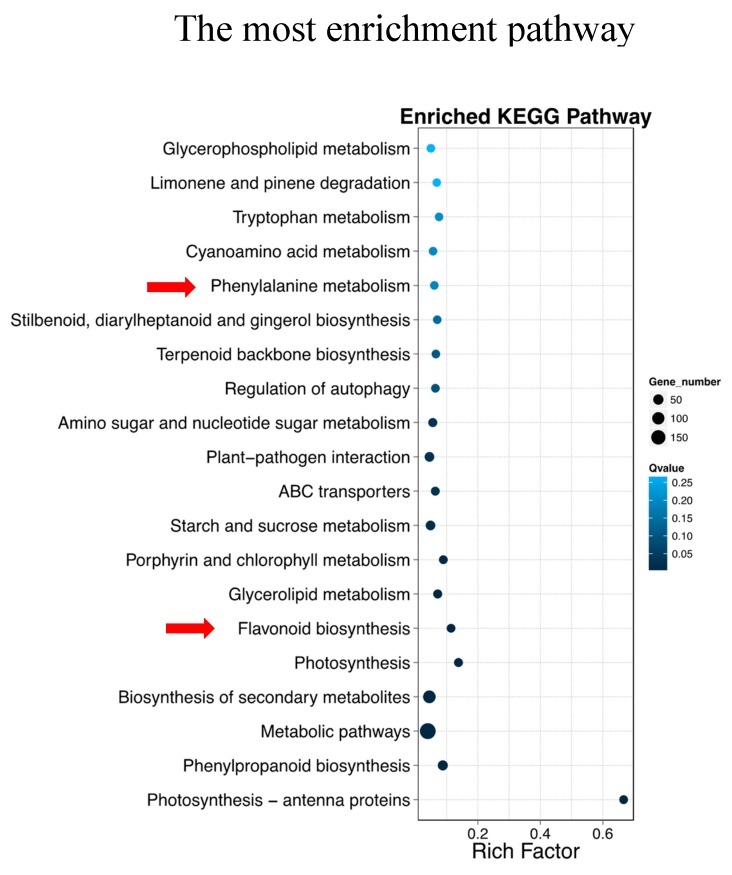
The most enrichment pathway of DEGs (TOP20). X axis: enrichment factor; Y axis: pathway name; The color: the q-value (high: white, low: blue), the lower q-value indicates the more significant enrichment; Point size: DEG number (The bigger dots refer to larger amount); Rich Factor: the value of enrichment factor, which is the quotient of foreground value (the number of DEGs) and background value (total Gene amount), the larger the value, the more significant enrichment. Red arrows represent the pathways related directly to the anthocyanin biosynthesis.

**Figure 4 ijms-20-04250-f004:**
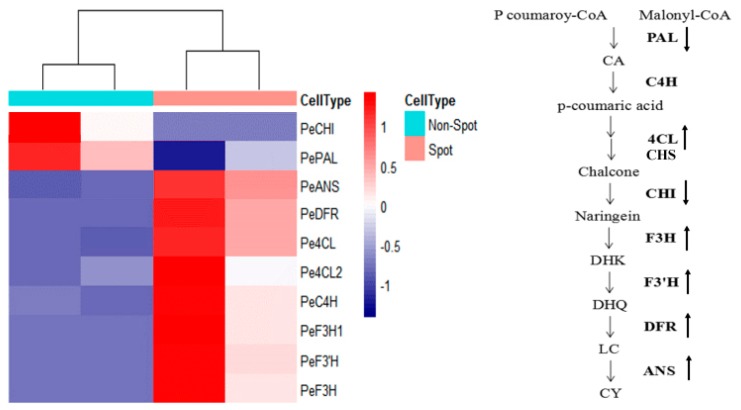
Heatmap of differentially expressed structure genes (DEGs) related to anthocyanin biosynthesis in *Phalaenopsis*. Bold arrow means up/downregulations of genes. (PAL, phenylalanine ammonia lyase; C4H, cinnamate-4-hydroxylase; 4CL, 4-coumarate--CoA ligase; CHS, chalcone synthase; CHI, chalcone isomerase; F3H, flavanone 3-hydroxylase; F3′H, Flavonoid 3’-hydroxylase; DFR, dihydroflavonol 4-reductase; ANS, Anthocyanidin synthase; DHK, dihydrokaempferol; DHQ, dihydroquercetin; DHM, dihydromyricetin; LC, leucocyanidin; Cy, cyaniding).

**Figure 5 ijms-20-04250-f005:**
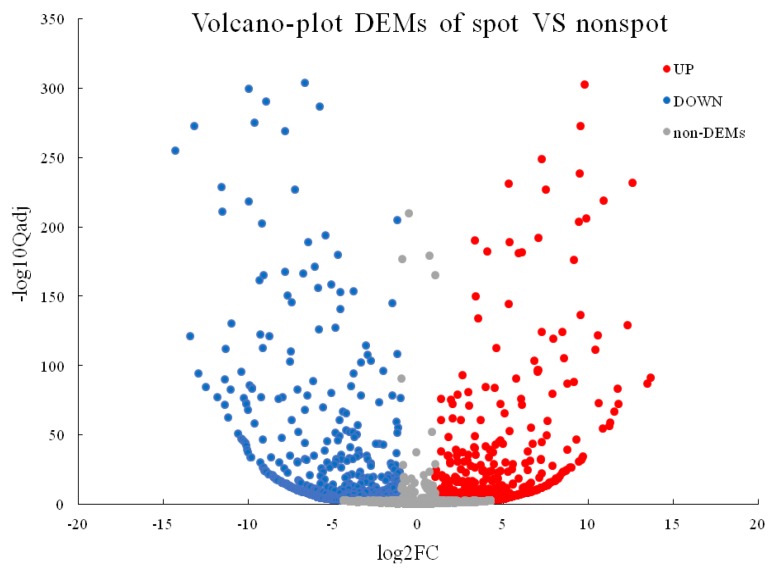
Volcano plot of DEMs. X axis represents log2 transformed fold change. Y axis represents −log10 transformed significance. Red points represent upregulated DEMs. Blue points represent downregulated DEMs. Gray points represent non-DEMs.

**Figure 6 ijms-20-04250-f006:**
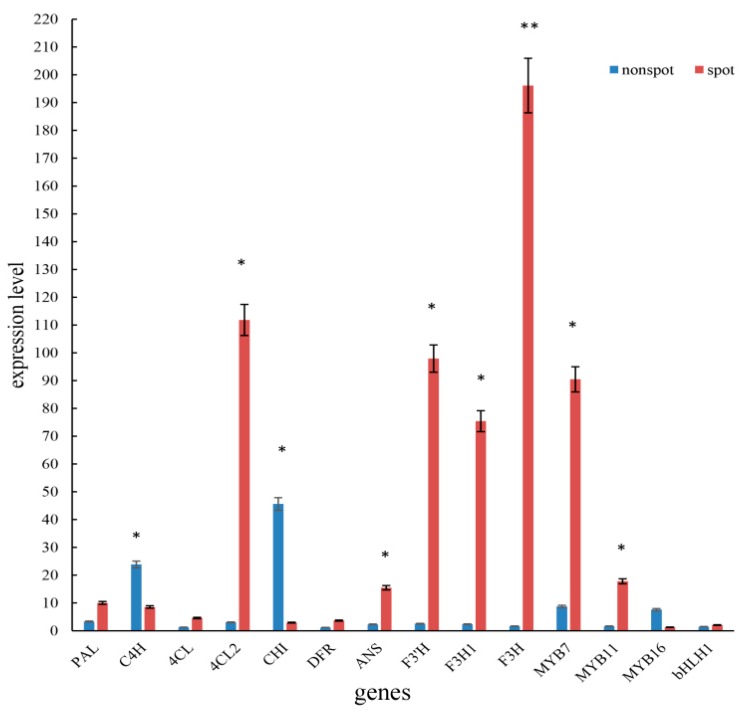
The expression level of structure gene and regulate gene Associated with Anthocyanin Biosynthesis, * represent there are significant differences between spot and non-spot area; ** represent there are extremely significant differences; red: spot area, blue: non-spot area.

**Figure 7 ijms-20-04250-f007:**
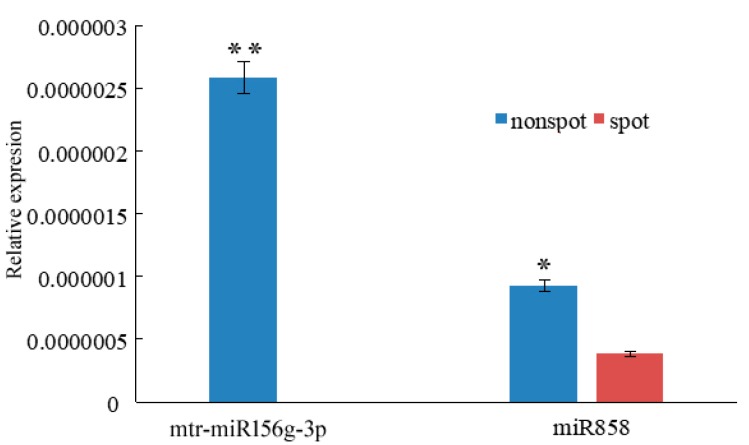
The expression level of mtr-miR156g-3p and miR858. ** represent an extremly significant difference (*p* < 0.01); * represent an significant difference (*p* < 0.05).

**Figure 8 ijms-20-04250-f008:**
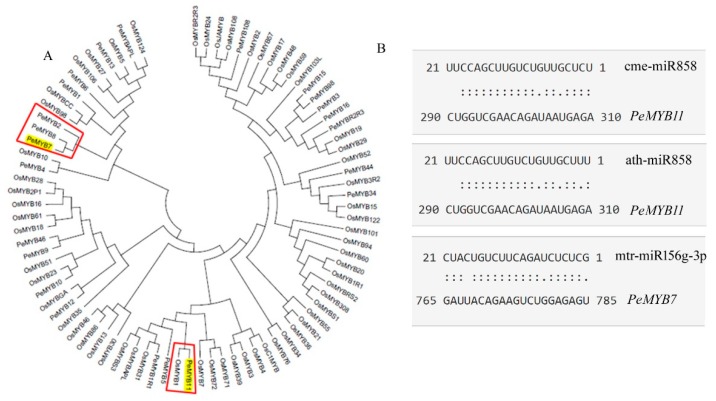
Identification of *PeMYB7*, *PeMYB11* and their target sites by miR156g, miR858. (**A**) Phylogenetic tree inferred from the MYB sequences of *Phalaenopsis equestris* and *Oryza sativa*; (**B**) Identification of target sites of miR156 and miR858 in *PeMYB7* and *PeMYB11.*

**Figure 9 ijms-20-04250-f009:**
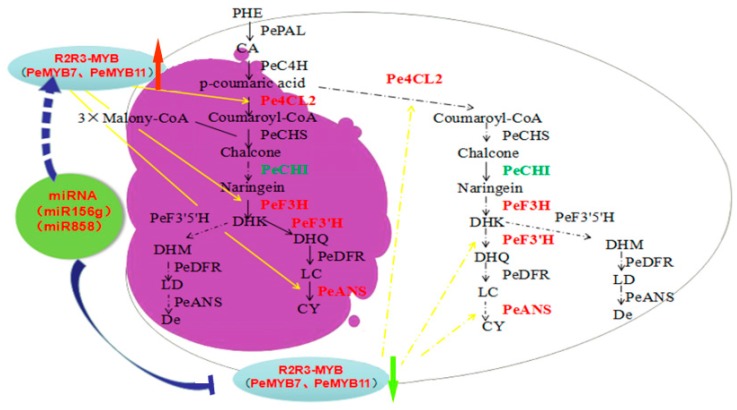
A proposed model summarizing of spot formation pattern in *Phalaenopsis* ‘Panda’ sepal. The purple segment illustrates the accumulation of anthocyanidin in spot area of ‘Panda’. MIR156g, MIR858 target on key regulate genes *PeMYB7* and *PeMYB11*, suppressed the transcript level of *Pe4CL2*, *PeF3H*, *PeF3′H*, *PeANS* and resulted in reduced anthocyanidin production in the non-spot area, while relatively lower expression levels of MIR156g and MIR858 and high levels of transcription of these genes in spot area cause accumulation of anthocyanidin. The black dashed arrows represent low levels of transcription, while the black solid-line arrows represent high levels of transcription. The yellow solid-line arrows represent promoting transcription, while yellow dashed arrow represent lack of promotion function. The blue bold solid-line T-arrow represents interfering transcription, while blue bold dashed arrow represents lack of interference function. The red bold arrow represents upregulation, and the green bold arrow represents downregulation. DHK, dihydrokaempferol; DHQ, dihydroquercetin; DHM, dihydromyricetin; LC, leucocyanidin; LD, leucodelphinidin; Cy, cyanidin; De, delphinidin.

**Figure 10 ijms-20-04250-f010:**
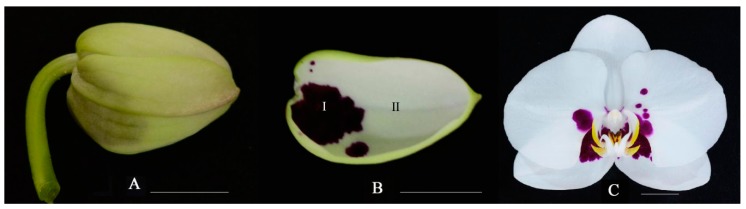
The flower tissue of *Phalaenopsis* ‘Panda’. Bars = 1 cm. (**A**) Full bud; (**B**) Sepal; I. spot area; II. non-spot area. (**C**) Full bloom stage.

**Table 1 ijms-20-04250-t001:** Putative anthocyanin structural genes identified from differentially expressed genes (DEGs).

Gene ID	Annotation	FPKM (non-spot)	FPKM (spot)	Log2FC	Padj	Up/Downregulation
PEQU_25924	*PeANS*	119.9303758	21,612.25078	7.493508409	4.71 × 10^−^^130^	Up
PEQU_00400	*PeF3′H*	31.02321	6117.42	7.623432	5.88 × 10^−^^88^	Up
PEQU_12025	*PeC4H*	2272.365	8325.431	1.87333	1.79 × 10^−^^8^	Up
PEQU_34933	*PeDFR*	17.40537	5973.723	8.422954	1.13 × 10^−^^117^	Up
PEQU_22606	*PeCHI*	2467.193	707.8831	−1.80129	1.13 × 10^−^^6^	Down
PEQU_07458	*Pe4CL*	11.87304	253.4331	4.415844	3.15 × 10^−^^18^	Up
PEQU_38891	*PeF3H1*	52.59874	5401.439	6.682172	2.59 × 10^−^^29^	Up
PEQU_22432	*PeF3H*	21.49244	3076.48	7.161307	2.17 × 10^−^^65^	Up
PEQU_01877	*PePAL*	14783.45	6368.878	−1.21487	8.83 × 10^−^^8^	Down
PEQU_00756	*Pe4CL2*	383.56	1290.653	1.750578	0.000153	Up

**Table 2 ijms-20-04250-t002:** Putative MBW genes identified from differentially expressed genes (DEGs).

Gene ID	Annotation	FPKM (non-spot)	FPKM (spot)	Log2FC	Padj	Up/Downregulation
PEQU_03393	*PeMYB7*	29.41538126	113.6604505	0.04965	0.00303	UP
PEQU_10361	*PeMYB11*	10.58835723	1693.628622	7.32149	8.71 × 10^−62^	UP
PEQU_10362	*PeMYB11*	4.49564678	216.34208328	5.58864183	2.35 × 10^−18^	UP
PEQU_09064	*PeMYB16*	1022.11354	508.0418102	−1.00853	0.03605	DOWN
PEQU_19747	*PebHLH1*	149.2074009	801.9464953	2.42618	2.57 × 10^−18^	UP

**Table 3 ijms-20-04250-t003:** Putative miRNAs targeting on *MYB*, *bHLH* or *WRKY* unigenes.

miRNA id	miRNA Expression in Non-spot	miRNA Expression in Spot	Up/Down Regulation of miRNA	*p*-Value	Target Gene	Target Unigene ID	Up/Downregulation of Target Unigene in Spot
Novel-m1700-5p	12	0	UP	0.000818887	PeMYB39	PEQU_22029	NA
mtr-miR156g-3p	49	1	UP	2.66 × 10^−12^	PeMYB7	PEQU_03393	UP
Novel-m0210-3p	82	10	UP	1.69 × 10^−13^	PeMYB7	PEQU_03393	UP
cme-miR858	52	0	UP	1.57 × 10^−14^	PeMYB11	PEQU_10361	UP
					PeMYB11	PEQU_10362	UP
					PeMYB8	PEQU_10866	NA
					PeMYB12	PEQU_20333	NA
ath-miR858	336	116	UP	4.67 × 10^−34^	PeMYB11	PEQU_10361	UP
					PeMYB11	PEQU_10362	UP
					PeMYB12	PEQU_20333	NA
ata-miR528-3p	12	0	UP	0.000818887	PebHLH086	PEQU_08299	NA
zma-miR528a-3p	49	1	UP	2.66 × 10^−12^	PebHLH086	PEQU_08299	NA
osa-miR162b	82	10	UP	1.69 × 10^−13^	PebHLH13	PEQU_33912	NA
Novel-m0112-3p	0	52	DOWN	1.57 × 10^−14^	PebHLH	PEQU_26133	NA
gma-miR169v	54	0	UP	2.83 × 10^−13^	katanin p80 WD40	PEQU_05516	NA
Novel-m0290-5p	24	0	UP	9.92 × 10^−7^	katanin p80 WD40	PEQU_05516	NA

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
