# Peer review of "mRNA and miRNA Expression Analysis Reveal the Regulation for Flower Spot Patterning in Phalaenopsis ‘Panda’"

_ijms, 2019, doi:10.3390/ijms20174250_

Round 1
Reviewer 1 Report
Dear Authors,
The subject of this paper is interesting for the Journal and for scientists. In my opinion, paper will be of interest to the international scientific community and deserves for publication in IJMS, although some changes are recommended.
Abstract:
Please change the second sentence “However, our knowledge of post-transcriptional regulation of anthocyanin biosynthesis by MYB genes and microRNA in Phalaenopsis is limited”. For someone who reads work for the first time, it is not clear why the authors indicate that participation of MYB and microRNA in regulation of anthocyanin biosynthesis is poorly understood. You provided information about this issue in the introduction. In my opinion in Abstract, should be only a general description of Ms in relation to post-transcriptional regulation.
Furthermore, there is no justification in the abstract why research was conducted on this species.
Introduction:
Are the data presented in the paper the first indicating that in Phalaenopsis ‘Panda’ flower spots are caused by the accumulation of anthocyanins? If there is such available data, please add proper references in the first paragraph.
Results:
The Fig. 1 (A, B, C) doesn’t contain any bars. On the Fig. 1D, E on the X axis there is no unit. Please add in the Fig. 1 some information, that the red color indicates the occurrence of anthocyanins.
The authors described “The expression levels of 10 anthocyanin genes and 4 transcriptor genes between the nonspot and spot areas were verified by qPCR. The results showed the genes Pe4CL2, PeCHI, PeANS, PeF3H, PeF3H1, PeF3’H and PeMYB7,PeMYB11 presented significantly higher expression levels in spot areas than the nonspot areas (Figure 6), which was generally consistent with the results of the transcriptome data (Table 1).” There was an error in the description of the results, the PeCHI gene expression is higher in nonspot (Fig. 6), the results in Table 1 indicate that it is down-regulated.
Fig. 6 – The signature of the X axis is not clear, please change to “gen”
Fig. 7 – A different gene name was used in the description of the X axis, another in the signature to the Fig, and another in the Table 3. Please clarify throughout.
Discussion:
Line 211: add information, that in Phalaenopsis ‘Panda’ expression of DFR in spot tissue was not significantly changed in comparison to nonspot area.
Inline 228 is written “In our study PeMYB7 and PeMYB11 expressed significantly different between the spot and nonspot areas, while PeMYB2 and PeMYB12 had not different expression levels (Table 2 and Fig.6).” In line 232 is written “The function of PeMYB7 is not very clear presently, however, in the phylogenetic tree inferred from MYB genes of Phalaenopsis equestris and Oryza sativa (Fig. 8A), PeMYB7 was in the same clad of PeMYB2 and PeMYB8, suggesting PeMYB7 may have similar function as PeMYB2 which can activate anthocyanin synthesis [43]. These two sentences are not logical, because if PeMYB2 not different expression levels then how can we say that PeMYB7, was in the same clad of PeMYB2 which can activate anthocyanin synthesis? Please clarify this issue.
Materials and methods
Plant materials
Line 294: at the end of the sentence insert reference to Fig. 10C. There is no information that the fresh material was collected for microscopic analyses. Which stage of flower development was selected for petals collection? Full bud, next full bloom stage?
In Fig. 10 change the order of the shots, first full bud, next sepal (if he was gathered from full bud), and full bloom. Please attach bars.
Line 303 - there is no information about the microscope equipment used for observation (name of the microscope, company).
Other
There are many grammatical and punctuation mistakes issues throughout the text that make it hard to understand what the authors intend to say. Especially in the Discussion section is sometimes hard to follow due to confusing grammar. Moreover, all paper contains are a lot of editorial errors, e.g. no spaces, unnecessary spaces, unnecessary commas, semicolons, etc. Please read the entire work carefully, including the captions for figures and tables, but also Reference list.
Author Response
Abstract:Please change the second sentence “However, our knowledge of post-transcriptional regulation of anthocyanin biosynthesis by MYB genes and microRNA in Phalaenopsis is limited”. For someone who reads work for the first time, it is not clear why the authors indicate that participation of MYB and microRNA in regulation of anthocyanin biosynthesis is poorly understood. You provided information about this issue in the introduction. In my opinion in Abstract, should be only a general description of Ms in relation to post-transcriptional regulation.
Furthermore, there is no justification in the abstract why research was conducted on this species.
Response: We have rewritten the abstract Line 19-26: Phalaenopsis cultivar ‘Panda’ is a beautiful and valuable ornamentals for its big flower and unique big spots on the petals and sepals. Although anthocyanins are konwn as the main pigments responsible for flower colors in Phalaenopsis and the anthocyanins biosynthetic pathway in Phalaenopsis is generally well known, the detailed knowledge of anthocynins regulation within the spot and nonspot parts in ‘Panda’ flower is limited. In this study, transcriptome and small RNA libraries analysis from spot and nonspot sepal tissues of ‘Panda’ were performed, and we found PeMYB7,PeMYB11 and miR156g, miR858 is associated with the purple spot patterning in its sepals.
Introduction:
Are the data presented in the paper the first indicating that in Phalaenopsis ‘Panda’ flower spots are caused by the accumulation of anthocyanins? If there is such available data, please add proper references in the first paragraph.
Response: ‘Panda’ is a cultivar of Phalaenopsis , and the references No.13 in second paragraph has indicated the main pigments in Phalaenopsis are anthocyanins. And in result, Figure 1 indicated that in Phalaenopsis ‘Panda’ flower spots are caused by the accumulation of anthocyanins (for the absorbance peak at 531nm).
Results:
The Fig. 1 (A, B, C) doesn’t contain any bars. On the Fig. 1D, E on the X axis there is no unit. Please add in the Fig. 1 some information, that the red color indicates the occurrence of anthocyanins.Response: The Fig.1 and its caption was fixed. (Line 110-114)
4.The authors described “The expression levels of 10 anthocyanin genes and 4 transcriptor genes between the nonspot and spot areas were verified by qPCR. The results showed the genes Pe4CL2, PeCHI, PeANS, PeF3H, PeF3H1, PeF3’H and PeMYB7,PeMYB11 presented significantly higher expression levels in spot areas than the nonspot areas (Figure 6), which was generally consistent with the results of the transcriptome data (Table 1).” There was an error in the description of the results, the PeCHI gene expression is higher in nonspot (Fig. 6), the results in Table 1 indicate that it is down-regulated.
Response: We have deleted PeCHI (Line 194).
5.Fig. 6 – The signature of the X axis is not clear, please change to “gen”
Response: We have changed the description of the X axis.
6.Fig. 7 – A different gene name was used in the description of the X axis, another in the signature to the Fig, and another in the Table 3. Please clarify throughout.
Response: We have changed the description of the X axis.
Discussion:
Line 211: add information, that in Phalaenopsis‘Panda’ expression of DFRin spot tissue was not significantly changed in comparison to nonspot area.Response: We have added “However, in Phalaenopsis ‘Panda’ expression of DFR in spot tissue was not significantly changed in comparison to nonspot area. “ (Line 217-218)
Inline 228 is written “In our study PeMYB7and PeMYB11expressed significantly different between the spot and nonspot areas, while PeMYB2 and PeMYB12 had not different expression levels (Table 2 and Fig.6).” In line 232 is written “The function of PeMYB7 is not very clear presently, however, in the phylogenetic tree inferred from MYB genes of Phalaenopsis equestrisand Oryza sativa (Fig. 8A), PeMYB7 was in the same clad of PeMYB2 and PeMYB8, suggesting PeMYB7 may have similar function as PeMYB2 which can activate anthocyanin synthesis [43]. These two sentences are not logical, because if PeMYB2 not different expression levels then how can we say that PeMYB7, was in the same clad of PeMYB2 which can activate anthocyanin synthesis? Please clarify this issue.Response: Since PeMYB7 function has not been studied in detail, and PeMYB7 and PeMYB8 are in the same branch, it is speculated that it may have the same function as PeMYB8. PeMYB8 has been shown to promote the synthesis of anthocyanidin. However, the two genes are not the same gene, and may have different expression and regulation patterns. Therefore,although the expression of PeMYB7 is significantly different, we cann’t judge that PeMYB8 should be also differentially expressed. This may be caused by the different gene expression among different cultivars.
Materials and methods
Plant materials
Line 294: at the end of the sentence insert reference to Fig. 10C. There is no information that the fresh material was collected for microscopic analyses. Which stage of flower development was selected for petals collection? Full bud, next full bloom stage?
Response: The fresh material were collected in the full bud. We have added the information in line 307 and 313.
In Fig. 10 change the order of the shots, first full bud, next sepal (if he was gathered from full bud), and full bloom. Please attach bars.
Response: The Fig.10 has been changed as the comment.
Line 303 - there is no information about the microscope equipment used for observation (name of the microscope, company).
Response: We have added theinformation about the microscope equipment used for observation. (Line 310)
Other
There are many grammatical and punctuation mistakes issues throughout the text that make it hard to understand what the authors intend to say. Especially in the Discussion section is sometimes hard to follow due to confusing grammar. Moreover, all paper contains are a lot of editorial errors, e.g. no spaces, unnecessary spaces, unnecessary commas, semicolons, etc. Please read the entire work carefully, including the captions for figures and tables, but also Reference list.
Response: We have corrected the grammatical and punctuation mistakes throughout the paper.
Reviewer 2 Report
In general the manuscript is ready to be published after very minor spill check, space and taking care of the following comments
Line 36-37: please change will improve on to will help in
Line 112 :- in the results please change 531 to 531 nm represents the absorption wavelength of anthocyanin
Line 114:- Please start the sentence with four instead of 4
Line 118:- add a space after 55.47%
Line 119:- change result to results to
Line 130-132:- Please adjust the space
Line:- please add a space after ,
Line 148:- take off the extra space after areas
Line 159:- take off the word there
Figure 7 need to be presented in a better way. The graph is very confusing
Is the miR 156g was presents in spot or nonspot. I think the comparison should be done between spot and nonspot as for the miR858
in Materials in Methods ;- please clarify the experimental designe ans state how many flowers or sepals the author originally started with and from how many individual plant?
In the supplemental table S1 please state what Q20% and Q30% present
Author Response
Comments:
Line 36-37: please change will improve on to will help in
Line 112 :- in the results please change 531 to 531 nm represents the absorption wavelength of anthocyanin
Line 114:- Please start the sentence with four instead of 4
Line 118:- add a space after 55.47%
Line 119:- change result to results to
Line 130-132:- Please adjust the space
Line:- please add a space after ,
Line 148:- take off the extra space after areas
Line 159:- take off the word there
Response: We have corrected the grammatical and punctuation mistakes according to these comments.
Figure 7 need to be presented in a better way. The graph is very confusing
Is the miR 156g was presents in spot or nonspot. I think the comparison should be done between spot and nonspot as for the miR858
Response: The expression level of miR 156g in nonspot was also done but not detected.
in Materials in Methods ;- please clarify the experimental designe ans state how many flowers or sepals the author originally started with and from how many individual plant?
Response: The biological replicates were presented in line 315.
In the supplemental table S1 please state what Q20% and Q30% present
Response: We have explained Q20% and Q30% in supplemental table S1.